

# Vertical profiles of aerosol mass concentrations observed during dust events by unmanned airborne in-situ and remote sensing instruments

Dimitra Mamali [1], Eleni Marinou [2,3], Jean Sciare [4], Michael Pikridas [4], Panagiotis Kokkalis [2,5], Michael Kottas [2], Ioannis Binietoglou [2,6], Alexandra Tsekeri [2], Christos Keleshis [4], Ronny Engelmann [7], Holger Baars [7], Albert Ansmann [7], Vassilis Amiridis [2], Herman Russchenberg [1], and George Biskos [4,1]

[1]Department of Geoscience and Remote Sensing, Delft University of Technology, Delft, The Netherlands
[2]Institute for Astronomy, Astrophysics, Space Applications and Remote Sensing, National Observatory of Athens, Greece
[3]Department of Physics, Aristotle University of Thessaloniki, Thessaloniki, Greece
[4]The Cyprus Institute, Energy, Environment and Water Research Centre, Nicosia, Cyprus
[5]Physics Department, Faculty of Science, Kuwait University, P.O. Box 5969, Safat 13060, Kuwait
[6]National Institute for Research and Development in Optoelectronics, Ilfov, Romania
[7]Leibniz Institute for Tropospheric Research, Leipzig, Germany

*Correspondence to:* Dimitra Mamali (D.Mamali@tudelft.nl) and George Biskos (G.Biskos@tudelft.nl; g.biskos@cyi.ac.cy)

**Abstract.** In-situ measurements using Unmanned Aerial Vehicles (UAVs) and remote sensing observations can independently provide dense vertically-resolved measurements of atmospheric aerosols; information which is highly required in climate models. In both cases, inverting the recorded signals to useful information requires assumptions and constraints, and this can make the comparison of the results difficult. Here we compare, for the first time, vertical profiles of the aerosol mass concentration

5  derived from Light Detection And Ranging (lidar) observations and in-situ measurements using an Optical Particle Counter (OPC) onboard a UAV during moderate and weak Saharan dust episodes. Agreement between the two measurement methods was within experimental uncertainty for the coarse-mode (i.e., particles having radii $> 0.5$ $\mu$m) where the properties of dust particles can be assumed with good accuracy. This result proves that the two techniques can be used interchangeably for determining the vertical profiles of the aerosol concentrations, bringing them a step closer towards their systematic exploitation in

10  climate models.

## 1  Introduction

Aerosol particles affect the atmospheric energy balance directly by interacting with solar radiation, and indirectly through the formation of clouds (Lohmann and Feichter, 2004). Determining the radiative forcing of the atmospheric aerosol particles is highly uncertain partly because of the significant spatial (both vertically and horizontally) and temporal variability of their con-



centration, size, and chemical composition (IPCC, 2013). The vertical variability in the properties of the atmospheric aerosol can be independently determined by modern in-situ measurements using airborne platforms and remote-sensing observations. Comparison of the measurements obtained by these two types of techniques, however, is instrumental for improving the accuracy of the resulting observational data for use in climate models.

Light Detection and Ranging (LIDAR) instruments are among the most powerful tools for probing vertically-resolved properties of the atmospheric aerosol. A number of retrieval algorithms that have been developed over the years can be used to obtain aerosol optical parameters from the lidar raw signals, including the aerosol backscatter coefficient $\beta_{aer}$ (Klett, 1981; Fernald, 1984), the aerosol extinction coefficient $\alpha_{aer}$ (Ansmann et al., 1990, 1992), and the particle depolarization ratio

$\delta^p$ (Freudenthaler et al., 2009). Under certain assumptions, recently developed algorithms can now be used to retrieve other vertically resolved aerosol properties such as particle absorption and mass concentration using the synergy of lidar and sunphotometer (Ansmann et al., 2011; Lopatin et al., 2013; Chaikovsky et al., 2016). To check the validity of these assumptions and to assure the quality of the final data, certain aerosol properties retrieved from lidar observations have been compared with vertical in-situ observations using research aircraft (Feingold and Morley, 2003; Weinzierl et al., 2011; Bravo-Aranda et al.,

2015; Granados-Muñoz et al., 2016; Rosati et al., 2016; Kokkalis et al., 2017; Tsekeri et al., 2017).

Airborne in-situ measurements using research aircraft are complex and costly, and therefore their availability is scarce and time-restricted, limiting comparability with remote sensing observations. What is more, manned aircraft cannot cover the lowermost part of the atmosphere due to safety restrictions, posing another major limitation. Recent efforts in aerosol in-

strumentation have provided lightweight and miniaturized instruments that can measure the size and concentration of aerosol particles onboard UAVs (Altstädter et al., 2015; Bezantakos et al., 2015; Barmpounis et al., 2016; Brady et al., 2016; Renard et al., 2016) in a much simpler and cost-effective manner. As a result, vertical profiling of key aerosol parameters can now be performed over long periods of time on a routine basis, and at much lower altitudes compared to measurements with manned research aircraft. Considering, however, that these advantages come in many cases at the expense of the quality of the recorded

data, measurements of aerosol properties using miniaturized instruments onboard UAVs need to be validated before using them to bridge the long-lasting gap between in-situ measurements and remote sensing observations.

Here we compare, for the first time to our knowledge, vertical profiles of the aerosol mass concentration, derived from lidar measurements using the POlarization LIdar PHOtometer Networking technique (POLIPHON), and in-situ measurements with

an OPC onboard a UAV (hereafter referred to as OPC$_a$). It should be noted that the two techniques do not measure the mass concentration directly, but this is estimated from the recorded signals of the two instruments. The measurements were recorded during the BACCHUS-INUIT-ACTRIS (Impact of Biogenic Versus Anthropogenic emissions on Clouds and Climate: towards a Holistic UnderStanding; Ice Nuclei Research Unit; European Research Infrastructure for the observation of Aerosol, Clouds and Trace gases Research InfraSctructure network) campaign that took place in Cyprus during April 2016.





## 2 Instrumentation and Methods

### 2.1 Site Description

Cyprus is located in the Eastern Mediterranean (cf. Figure 1 inset), receiving air masses from Europe, Middle East and North Africa (Lelieveld et al., 2002). Therefore, it represents an ideal location for characterizing different aerosol types and investi-

5 gating the role of particles in various atmospheric processes.

The measurements reported here were conducted at three different locations. Aerial measurements, using a UAV, were carried out at Orounda ($35^o09'$ N; $33^o07'$ E; 310 m above sea level; a.s.l.) providing highly-resolved spatially and temporally distributed data up to ca. 2 km above ground level (a.g.l). Ground-based in-situ aerosol measurements, were performed at the Cyprus Atmospheric Observatory (CAO) at Agia Marina-Xyliatou ($35^o04'$ N; $33^o06'$ E; 535 m a.s.l.), located 6.5 km south of

10 Orounda. A Polly$^{XT}$ Raman lidar was located at the suburbs of Nicosia ($35^o14'$ N; $33^o38'$ E; 190 m a.s.l.), ca. 35 km east of Orounda, providing round-the-clock measurements of the atmospheric conditions up to 12 km a.g.l.. The exact locations of the measuring points are shown in Figure 1 and detailed descriptions of the instruments are given below.

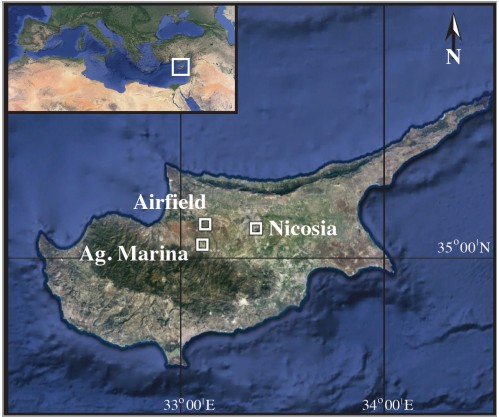

**Figure 1.** Map of of Cyprus showing the locations of the observation sites used for the measurements reported in this paper. The inset in the upper-left corner shows the greater area of South Europe, North Africa, and the Middle East, with the white square showing the location of Cyprus. The maps were generated with Google Earth Pro version 7.1.7.2606 (https://www.google.com/earth/download/gep/agree.html).

### 2.2 Unmanned Aerial Vehicle (UAV)

The UAV employed during the campaign (Figure 2) has a fixed wingspan of 3.8 m, and is powered by a two-stroke internal

15 combustion engine. It has a take-off weight of 35 kg that results in a payload capacity of approximately 12 kg. The payload bay is 1.3 m × 0.23 m × 0.34 m (length-width-height), and can fit multiple instruments. When loaded, the UAV can fly for up to 4 hours with an air speed velocity of $25 \pm 10$ m s$^{-1}$ and can reach altitudes up to 4 km a.g.l. (due to airspace limitations, however, only flights up to 2 km were permitted). An autopilot system allowed predetermined flight plans that involved spiral rectangular-shaped ascending and descending patterns (cf. Figure S1 of the Supplementary Material) preventing contamination





of the sampling system from the engine's exhaust. For consistency, the results shown in the rest of the paper correspond to measurements during ascends.

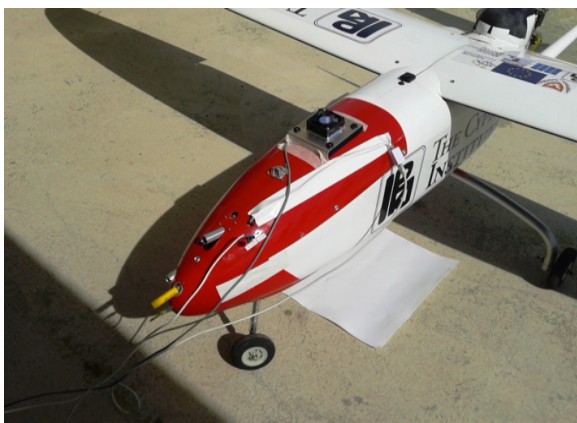

**Figure 2.** Photograph of the UAV of the Cyprus Institute used for the measurements reported in this work.

### 2.3 UAV-based Optical Particle Counter Measurements

Vertical profiles of the particle size distributions of the atmospheric aerosol were measured using an OPC (MetOne, Model
212-2) onboard the UAV. The sampled aerosol was dried to below 50% relative humidity (RH) by gently heating the sampling tube of $OPC_a$, which reported particle size distributions, ranging from 0.15 to 5 $\mu$m in radius, in 8 size bins. Assuming spherical shape and constant mass density for the particles, the size distributions were converted into aerosol mass concentrations (see Section S1 in the Supplementary Information). In addition to $OPC_a$, a single wavelength aethalometer (AethLabs - Model AE51) was onboard the UAV to verify that no contamination of the sampled air by the engine exhaust took place. Each instru-
ment was equipped with an individual sampling inlet that extended 5 cm from the UAV nose to ensure representative sampling.

### 2.4 Lidar Measurements

A depolarization Raman lidar Polly$^{XT}$ (Althausen et al., 2009; Engelmann et al., 2016) was used in the measurements reported here. This lidar emits laser pulses simultaneously at three wavelengths: 1064, 532 and 355 nm. The laser beam interacts with
the atmospheric molecules and particles, and a part of it (backscattered light) is collected by the receiver unit that consists of two telescopes (near-field and far-field). The elastically backscattered signals are used as input to the Fernald-Klett algorithm (Klett, 1981; Fernald, 1984; Böckmann et al., 2004) to retrieve the vertical profile of the particle backscatter coefficient $\beta_{aer}$. This method assumes a linear relationship between the aerosol extinction-to-backscatter ratio constant (i.e., the lidar ratio S) throughout the entire atmospheric column; a critical assumption that can induce uncertainties up to 20-30% of the retrieved
property from statistical and systematic errors (Bösenberg and Brassington, 1997; Comerón et al., 2004; Rocadenbosch et al.,



2010).

In addition to the elastically backscattered signal, Polly$^{XT}$ receives the nitrogen Raman-shifted signal at wavelengths 387 and 607 nm, and the water vapor Raman signal at 407 nm wavelength. The Raman technique (Ansmann et al., 1992; White-
man et al., 1992) utilizes the elastic and inelastic signals to retrieve the particle extinction $\alpha_{aer}$ and scattering $\beta_{aer}$ profiles independently, without any critical assumptions. The range-resolved aerosol lidar ratio can then be directly estimated as the ratio $\alpha_{aer}/\beta_{aer}$. In our analysis, we used the Raman technique to retrieve the $\alpha_{aer}$ and $\beta_{aer}$ profiles during night-time, and the Fernald-Klett method during day-time when the Raman signal is highly affected from the background noise induced by the scattered sunlight. The Polly$^{XT}$ system also provides information on volume depolarization ratio $\delta^v$ from which the par-
ticle depolarization ratio $\delta^p$ can be estimated (Murayama et al., 1999; Sakai et al., 2000; Shimizu et al., 2004; Sugimoto and Lee, 2006; Freudenthaler et al., 2009). This allows discrimination between spherical particles (e.g., water droplets) and non-spherical particles such as dust.

## 2.5 Sun/sky Photometer Measurements

A lunar/sun sky photometer of the Aerosol Robotic Network (AERONET; Holben et al. 1998) was collocated with the lidar at Nicosia, whereas an additional sunphotometer was situated at CAO. Both instruments provided measurements of the Aerosol Optical Thickness (AOT) at seven wavelengths (i.e., 340, 380, 440, 500, 675, 871 and 1020 nm). The AERONET products include, among others, parameters corresponding to the total atmospheric column such as the Ångström exponent Å (at several wavelength pairs), the particle volume size distributions in the size range 0.05 to 15 $\mu$m (particle radius), the fine- and
coarse-mode AOT ($\tau_f$ and $\tau_c$, respectively) at 440, 675, 871, 1020 nm (O'Neill et al., 2003) and the fine- and coarse-mode volume concentrations ($v_f$ and $v_c$, respectively; Dubovik et al., 2000, 2006). Cloud screened and quality assured level 2.0 data products were used in this work. The uncertainties of the AOT were $< 0.02$ for UV wavelengths and $< 0.01$ for wavelengths above 440 nm (Eck et al., 1999).

## 2.6 Mass Concentration Profiles - The POLIPHON Method

The mass concentration profiles from the lidar measurements were calculated using the POLIPHON method (Ansmann et al., 2011) as stated above. In the first step of the method, the contribution of the fine-mode and coarse-mode particles to the total backscatter coefficient ($\beta_t$) is calculated based on depolarization measurements (Tesche et al., 2009a). Here we assumed an externally-mixed aerosol consisting of a fine component with low depolarization ($5 \pm 1\%$; Ansmann et al., 2011), and a coarse
component that induces light depolarization of $31 \pm 4\%$ (Freudenthaler et al., 2009), corresponding to dust particles. The





dust-related backscatter coefficient was determined as:

$$\beta_d = \beta_t \frac{(\delta_t - \delta_{nd})(1 + \delta_d)}{(\delta_d - \delta_{nd})(1 + \delta_t)} \quad , \tag{1}$$

where $\delta_t$, $\delta_{nd}$ and $\delta_d$ are respectively the observed total depolarization ratio, the assumed non-dust depolarization ratio and the measured depolarization ratio of dust particles. Once $\beta_d$ was determined, the non-dust backscatter coefficient was calculated

by $\beta_{nd} = \beta_t - \beta_d$. In the calculations presented here we used $\beta$ and $\delta^p$ values corresponding to 532 nm wavelength.

In the second step of the method, the mass concentrations of the fine (non-dust; $m_{nd}$) and coarse (dust; $m_d$) aerosol fractions are calculated according to (Ansmann et al., 2011):

$$m_d = \rho_d \overline{(v_c/\tau_c)} \beta_d S_d \tag{2}$$

$$m_{nd} = \rho_{nd} \overline{(v_f/\tau_f)} \beta_{nd} S_{nd} \quad , \tag{3}$$

where $\rho$ is the mass density, whereas the product of the backscattering coefficient and the lidar ratio $\beta \cdot S$ is the extinction coefficient of the particles, with subscripts $d$ and $nd$ denoting dust (coarse) and non-dust (fine) particles. It should be noted that the factors $\overline{v_c/\tau_c}$ and $\overline{v_f/\tau_f}$ are used to convert the extinction measurements to particle volume concentration for the coarse and the fine faction, respectively. In this work these factors were determined from the daily mean data of the sunphotometer that was collocated with the lidar. The volume concentrations $v_f$ and $v_c$ were obtained from the AERONET data, whereas the

fine and coarse mode AOTs, $\tau_f$ and $\tau_c$, at 532 nm wavelength, were calculated using Å according to:

$$\tau_{f,c(532)} = \tau_{f,c(440)} \times \left(\frac{440}{532}\right)^{\text{Å}_{f,c(675-440)}} \tag{4}$$

Another assumption we made was that the lidar-derived dust and non-dust fractions are identical to the photometer-derived coarse and fine particle fractions. The inflection point of the AERONET data was adopted as the limit between the fine and the

coarse-mode particles. As a result, the fine mode ranged between 0.05-0.5 $\mu$m (particle radius) and the coarse-mode between 0.5-15 $\mu$m as shown in Figure 3. The calculated values of $\overline{v_f/\tau_f}$ and $\overline{v_c/\tau_c}$ (cf. Table 1) are in line with the conversion factors mentioned by Mamouri and Ansmann (2016, 2017) who performed an extensive analysis of the conversion factors of dust over Cyprus.

Apart from $v/\tau$, the other parameters required for determining the aerosol mass concentration from the lidar measurements are $\rho$, $\beta$ and S. Regarding $\rho$, we used a density of 1.5 $\pm$ 0.3 g cm$^{-3}$ (Hess et al., 1998) for the fine-mode particles and 2.6 $\pm$ 0.6 g cm$^{-3}$ for the coarse-mode (corresponding to dust according to Gasteiger et al., 2011). Chemical analysis of filter samples collected during the measurements showed that the dust density assumed here is valid (data not shown). Values for S$_{nd}$ (60 $\pm$ 10 sr) were taken from the literature (Mamouri and Ansmann, 2014), and actual measurements were used for S$_d$. More




specifically, the $S_d$ value ($47 \pm 10$ sr) was estimated by night-time Raman measurements when pure and dense Saharan dust layers occurred over Nicosia on 15 April 2016 (cf. Figure S2 of the Supplementary Material). The analysis of the lidar data for the estimation of the $S_d$ along with the error calculations for equations (1) to (3) are given in the Supplementary Information (cf. Sections S2 and S3 in the Supporting Information). The mean uncertainties of $\beta_d$ and $\beta_{nd}$ were 22% and 28%, respectively.

5    All the values of the parameters that are required as input for the calculations are summarized in Table 1 .

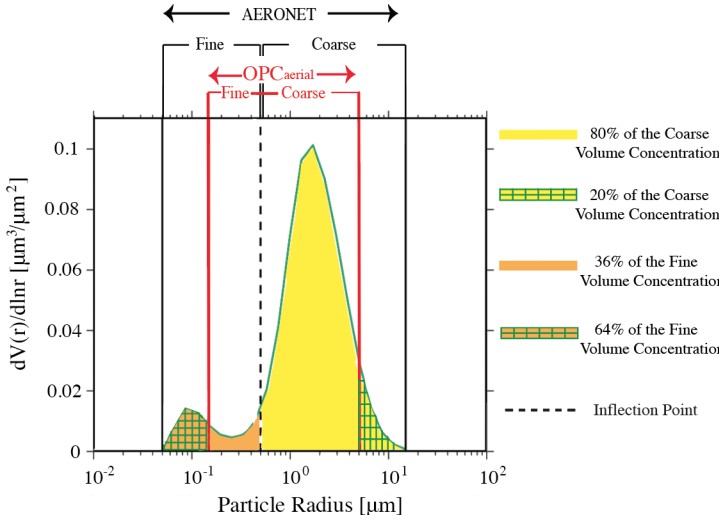

**Figure 3.** Column-integrated volume size distribution measured with the sunphotometer over Nicosia at 06:57 UTC on 15 April 2016. The ranges of particle sizes measured by AERONET sunphotometers, and the $OPC_a$ are also indicated in the figure.

**Table 1.** Values of input parameters used in the POLIPHON algorithm.

| Parameter | Symbol | Values | Source/Reference |
|---|---|---|---|
| Dust depolarization ratio | $\delta_d$ | $31 \pm 4\%$ | (Freudenthaler et al., 2009) |
| Non-dust depolarization ratio | $\delta_{nd}$ | $5 \pm 1\%$ | (Ansmann et al., 2011) |
| Dust lidar ratio | $S_d$ | $47 \pm 10$ sr | Raman measurements, this study |
| Non-dust lidar ratio | $S_{nd}$ | $60 \pm 10$ sr | (Mamouri and Ansmann, 2014) |
| Dust particle density | $\rho_d$ | $2.6 \pm 0.6\ g\ cm^{-3}$ | (Hess et al., 1998) |
| Non-dust particle density | $\rho_{nd}$ | $1.5 \pm 0.3\ g\ cm^{-3}$ | (Hess et al., 1998) |
| Dust conversion factor (15.04.2016) | $\overline{v_c/\tau_c}$ | $0.67 \pm 0.05 \times 10^{-6}$ | Sunphotometer, this study |
| Non-dust conversion factor (15.04.2016) | $\overline{v_f/\tau_f}$ | $0.24 \pm 0.018 \times 10^{-6}$ | Sunphotometer, this study |
| Dust conversion factor (22.04.2016) | $\overline{v_c/\tau_c}$ | $0.81 \pm 0.04 \times 10^{-6}$ | Sunphotometer, this study |
| Non-dust conversion factor (22.04.2016) | $\overline{v_f/\tau_f}$ | $0.14 \pm 0.019 \times 10^{-6}$ | Sunphotometer, this study |



## 3  Results and Discussion

### 3.1  Homogeneity of Aerosol Measurements over the Measurement Sites

Given the proximity (6.5 km) of the ground (at CAO) and the airborne in-situ observations (at Orounda), as well as the absence of any strong pollution sources in the region, the measurements were considered to correspond to the same air parcel in terms of atmospheric composition. The third measurement location (Nicosia) was situated 35 km away from the airfield. As suggested from the comparison of sunphotometer measurements at Nicosia and CAO, however, all locations were influenced by the same air masses with minor local influence which was mostly trapped within the PBL.

Figure 4 shows the $AOT_{500}$ and the $Å_{440-870}$ measured by the sunphotometers in Nicosia and at CAO from 13 to 24 April 2016 when concurrent measurements were performed at the two locations. Overall, the temporal variability of these two parameters observed at Nicosia was very similar with the respective measurements at CAO, exhibiting correlations coefficients of 0.89 and 0.87 for $AOT_{500}$ and $Å_{440-870}$, respectively. This good correlation was further enhanced during the dust event cases (e.g. on 15 April 2016), when the relative contribution of the aerosol fine mode was minimized. In terms of absolute values, $AOT_{500}$ was 15-50% higher at Nicosia compared to CAO, even during the cases with the dust events, when coarse particles dominated. These higher values at Nicosia are mainly due to the altitudinal difference between the sites (Nicosia is at an altitude of 190 m whereas CAO at 535 m) and the contribution of the local aerosol sources to the total aerosol burden. This was further justified by the higher $Å_{440-870}$ measurements at Nicosia which signify the presence of small aerosol particles from anthropogenic sources.

### 3.2  Comparison of the Mass Concentration Measurements

A total of 6 UAV flights with $OPC_a$ onboard were performed during the entire campaign. However, only 2 fulfilled all the necessary requirements for comparison with the lidar observations. Those requirements are that 1) there are simultaneous measurements of lidar and $OPC_a$, 2) there are cloud-free conditions or clouds are above 7-8 km altitude so that the lidar retrievals can be made, 3) there is enough dust loading, 4) there is availability of AERONET data and 5) the airborne in-situ measurements were performed before the full development of the PBL. All these requirements were fulfilled during the measurements on 15 April 2016 and 22 April 2016, which are analyzed below.

#### 3.2.1  Case Study I: 15 April 2016

The atmospheric situation over South-Eastern Europe on 15 April 2016 was dominated by a high-pressure system resulting in mostly cloud-free conditions over Cyprus. A dust event of moderate intensity was observed, resulting in an average $AOT_{500}$ value of 0.4 over Nisosia and CAO (cf. discussion in section 3.1 and Figure 4). Figure 5 shows the lidar time-height display during that day, with: the upper panel showing the range-corrected signal of the 1064-nm channel, which provides information



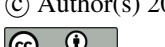

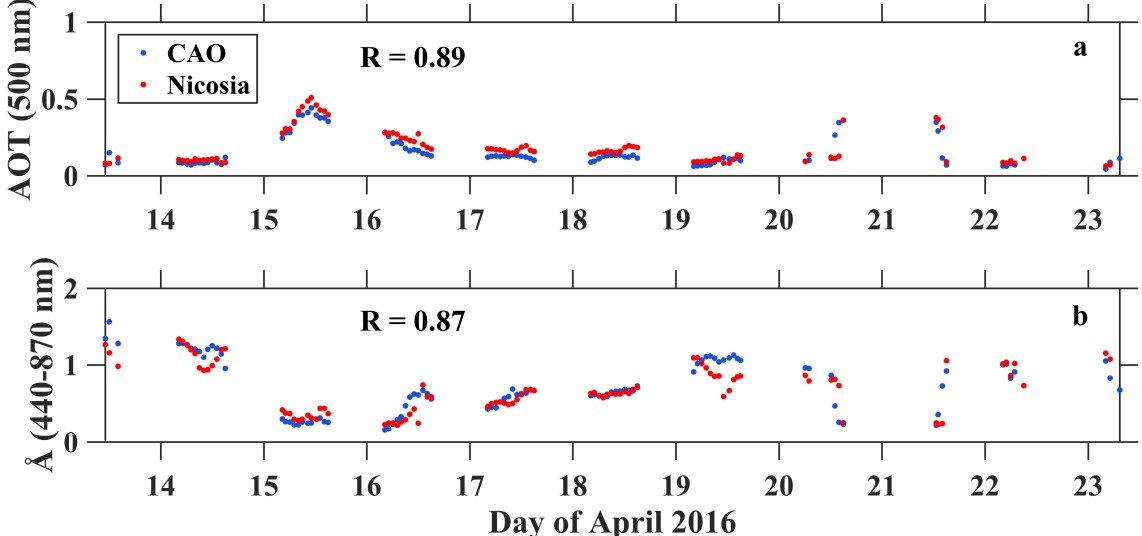

**Figure 4.** (a) $AOT_{500}$ and (b) $Å_{440-870}$ as measured with the sun-photometers at CAO (blue circles) and Nicosia (red circles) from 13 to 24 April 2016.

about the aerosol loading and the presence of clouds, and the lower panel the linear volume depolarization ratio $\delta^v$ at 532 nm that can be used to discriminate particles of different shapes that can be indicative of different sources. Throughout the day, high concentrations of aerosol particles were observed even up to ca. 7 km altitude (Figure 5a), with a persistent aerosol layer extending from 2.5 to ca. 7 km. Backtrajectory analysis (cf. Figure 6) corroborated that this layer resulted from a Saharan dust

5 event that originated in Algeria and traveled over Greece and Turkey before reaching Cyprus. The $\delta^v$ plot (Figure 5b) also shows the temporal evolution of this dust layer. From 00:00-03:00 UTC the dust extends from 2 to 7 km altitude, but later (until 14:00 UTC) it becomes shallower. From the early morning hours (07:00 UTC) to early afternoon (14:00 UTC) when the boundary layer develops, the dust layer is confined above it, reaching an altitude of up to 5 km. After the collapse of the boundary layer, the dust layer starts to descend and finally reaches the ground at 18:00 UTC.

The cloud-free and time homogeneous atmospheric scene between 07:00 and 07:50 UTC, which overlapped with the time window of the UAV flight, was selected for calculating the parameters of the atmospheric aerosol using the POLIPHON method. The lidar profiles of $\beta$ (355 nm, 532 nm, 1064 nm; retrieved with the Fernald-Klett method (Section 2.4) and $\delta^p$ (355 nm, 532 nm), that were used as input in POLIPHON, are shown in Figure 7a-b. The backscatter signal increased gradually

15 from 1 $Mm^{-1}sr^{-1}$ at 1 km (532 nm), reaching a maximum of ca. 2.3 $Mm^{-1}sr^{-1}$ at 3 km where the dust layer core was. The pure dust layer spanned from ca. 2.5 to 3.8 km ($\delta^p \approx 30\pm2\%$) while below 2 km, the dust was mixed with almost spherical particles, probably from the residual layer, as indicated by the relatively low $\delta^p$ values ranging between 12% and 30%. Figure 7c shows the POLIPHON-derived dust and non-dust related backscatter coefficients $\beta_d$ and $\beta_{nd}$ derived by Equation (1), and respective uncertainties determined by Monte-Carlo calculations (cf. Section S2 in the Supplementary Information for details).





The backscatter coefficient of the fine-mode particles $\beta_{nd}$ decreased with altitude, while the dust particles were present even down to 0.7 km. As discussed in 2.4, the lidar ratio value used in the Fernald-Klett retrieval and the the lidar ratio corresponding to the dust particles $S_d$ that is required as input in the POLIPHON algorithm, were estimated from Raman lidar measurements that were performed between 00:00-01:40 UTC (UTC+3 local time), just before sunrise. It should be noted here that Raman

5 measurements can only be performed during the night because during the day the scattered sunlight induces high background noise signal. The fact that the dust layer observed during the Raman measurements had the same origin and followed the same atmospheric path before reaching the measurement site between 07:00-07:50 UTC was confirmed by back-trajectory analysis (data not shown).

10 Vertical profiles of the RH measured with the UAV and predicted by the WRF−ARW atmospheric model (Skamarock and Klemp, 2008) showed that the atmosphere was dry enough (RH $\lesssim$ 50%), at the ground level and up to 4 km altitude (Figure 7d). As a result we could safely assume that the aerosol particles were dry and thus changes in the mass density and backscatter coefficient due to water uptake were negligible.

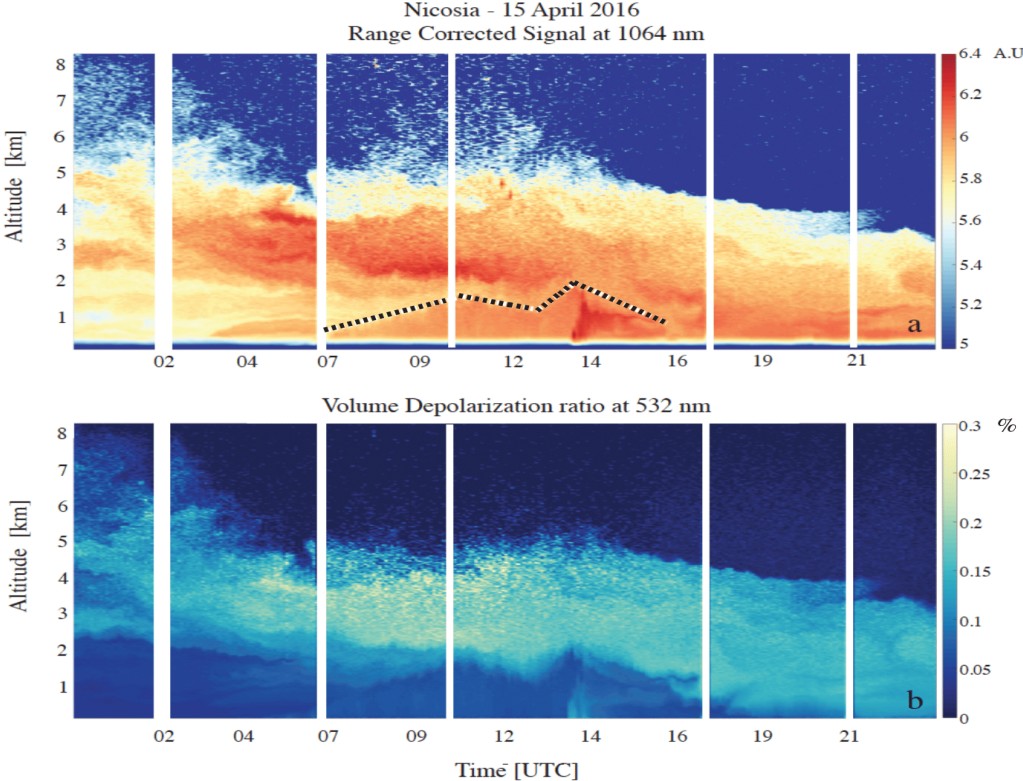

**Figure 5.** Range-corrected lidar signal at 1064 nm (a) and Volume Linear Depolarization ratio (b) reflecting the atmospheric conditions over Nicosia on 15 April 2016. Blue color indicates weak backscattering, yellow-red colors in the range corrected lidar signal (a) indicate backscattering mainly from fine aerosols and dust, whereas the dotted line shows the Planetary Boundary Layer (PBL) top. The lidar observations used for the comparison with the UAV measurements were those recorded between 07:00-07:50 UTC.





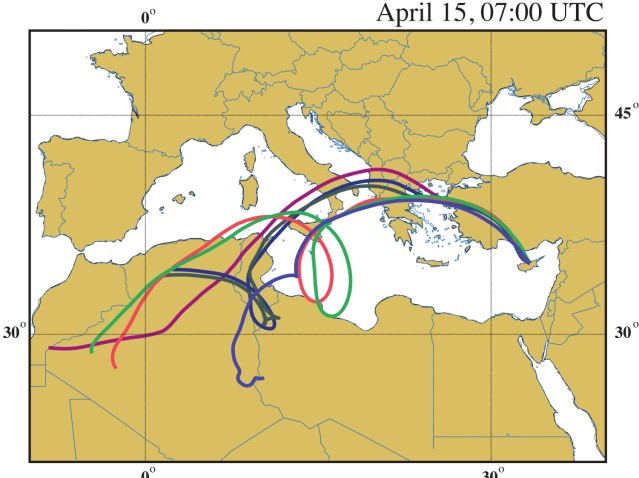

**Figure 6.** Back-trajectories of the air masses arriving at several altitudes over Cyprus on April 15, 07:00 (UTC). The back-trajectories were calculated for a duration of 5 days using the HYSPLIT transport and dispersion model (Rolph, 2003; Stein et al., 2015) with GDAS 1° meteorological data through the Real-time Environmental Applications and Display sYstem (READY; http://ready.arl.noaa.gov/index.php).

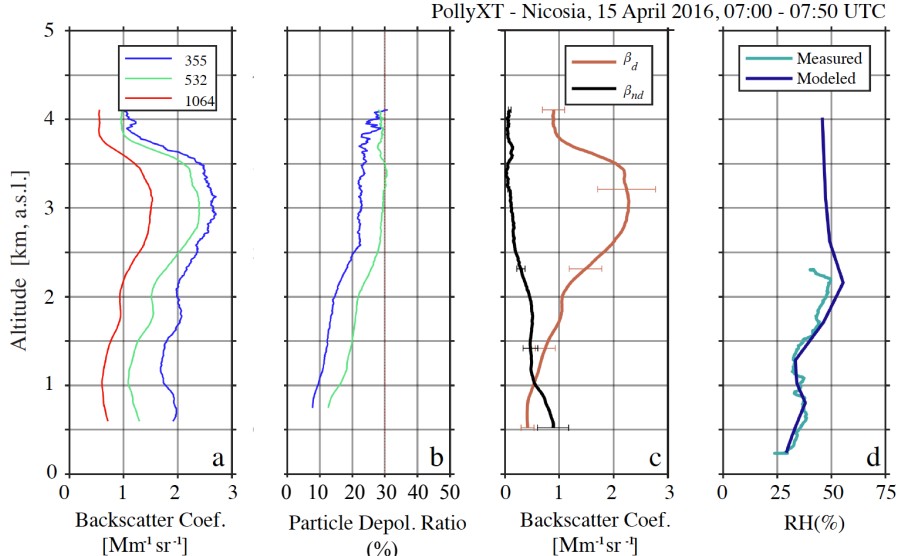

**Figure 7.** Daytime profiles of $\beta$ (355-, 532-, and 1064-nm wavelength) (a) $\delta^p$ (355- and 532-nm wavelength) (b)$\beta_d$ and $\beta_{nd}$ (c) determined by POLIPHON, as well as RH profiles from in-situ measurements onboard the UAV and from WRF−ARW model simulations over Nicosia at 08:00 UTC (d).





### 3.2.2 Case Study II: 22 April 2016

Contrary to case I, a low intensity dust event ($AOT_{500}$ = 0.1) was recorder over Cyprus on 22 April 2016. The evolution of the boundary layer dominating the atmospheric situation that day is depicted in the lidar time-height plots shown in Figure 8. From 00:00 to 10:00 UTC a sparse dust layer extended between 1 and 2 km a.g.l while after the PBL decay a shallower dust

5   plume was observed between 1 and 1.5 km altitude. According to the back-trajectory analysis (Figure 9) the dust air mass at 1.5 km originated from Egypt at the ground level, then it was elevated and passed over Libya, the Mediterranean and Turkey before reaching Cyprus.

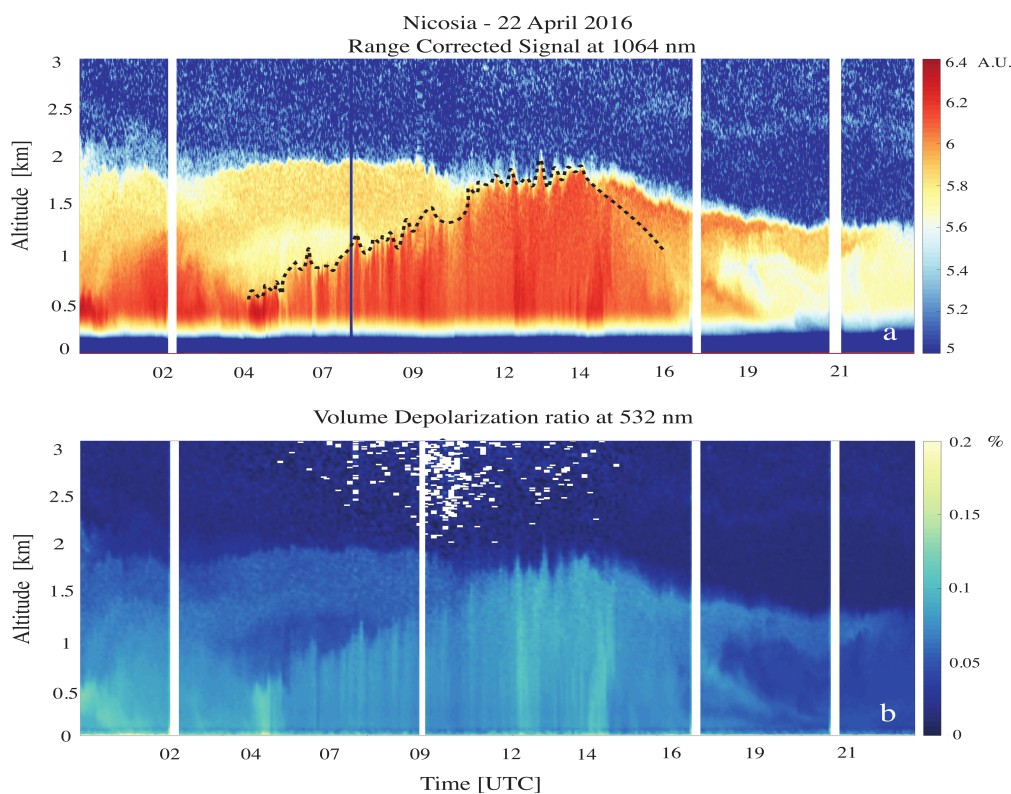

**Figure 8.** Range-corrected lidar signal at 1064 nm (a) and Volume Linear Depolarization ratio (b) reflecting the atmospheric conditions over Nicosia on 22 April 2016. In (a), blue color indicates weak backscattering, yellow-red colors indicate backscattering mainly from fine aerosols and dust. The dotted line shows the Planetary Boundary Layer (PBL) top. The lidar observations used for the comparison with the UAV measurements were those recorded between 04:22-05:00 UTC.

The UAV flight on that day was performed between 04:22 and 05:16 UTC. The atmospheric scene between 04:20-05:00

10   UTC (Figure 8) was chosen for the comparison due to its stable conditions above 0.8 km. Also in this case, the same procedure as in case I, was followed to retrieve the lidar profiles that were used as input in the POLIPHON algorithm. The backscatter coefficient, the particle depolarization ratio, the POLIPHON-derived dust and non-dust related backscatter coefficients as well





as the RH profiles of this atmospheric scene are shown in Figure 10. In contrast to the estimated $\delta^p$ values determined from the measurements on 15 April, here $\delta^p_{532}$ decreases gradually with height from 0.8 to 2 km having values between 10-17%. The combination of these relatively low $\delta^p_{532}$ values with the estimated $S_{532}$ value of $\approx 30$ Sr (estimated from Raman retrievals between 01:00-03:00 UTC) indicate a mixture of non-spherical dust particles with almost spherical aerosols, possibly of marine origin (Amiridis et al., 2005; Tesche et al., 2009b).

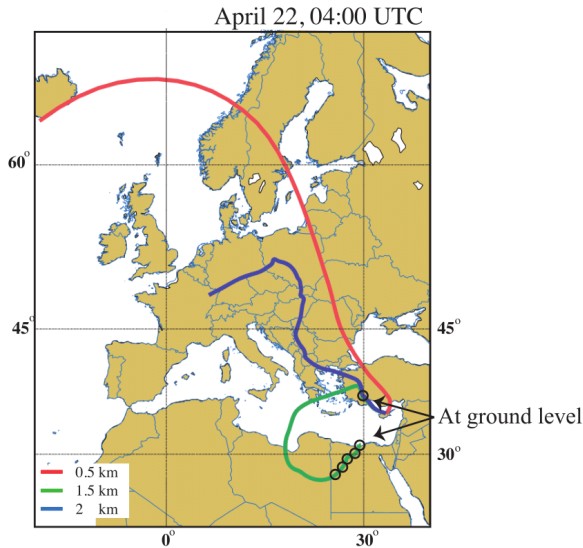

**Figure 9.** Back-trajectories of the air masses arriving at 500 m, 1000 m and 1500 m over Cyprus on April 22, 04:00 (UTC). The back-trajectories were calculated for a duration of 6 days; the black circles indicate the locations where the air-mass was below 100 m altitude.

### 3.2.3 Mass Concentration Profiles

Figure 11 (a,c) show the mass concentration profiles for the coarse particles (particles larger than 0.5 $\mu$m in radius) as derived by the lidar observations using POLIPHON method for inversion, and the $OPC_a$ measurements. The lidar profiles, were calculated by Equations (2) and (3) using the measured $\beta_d$ and $\beta_{nd}$, profiles and the dust density values from the literature (cf. Table 1). The respective $OPC_a$ profiles were determined by the recorded particle number size distributions assuming the same dust particle density (cf. Section 2.3 for details). To ensure that the lidar observations are representative of the atmospheric aerosol over Orounda and over CAO we compare the data for altitudes higher than 0.8 km a.s.l. during morning hours when the PBL was shallow.

The mass concentration profiles from the lidar and the $OPC_a$ observed on 15 April 2016 (Figure 11a), show a good correlation, with $R^2 = 0.8$. In terms of absolute values, the mass concentrations measured by the $OPC_a$ (red curve) lie within the uncertainty limits (38%) of the lidar observations, with the former being equal or lower for the entire range of altitudes, exhibiting a bias ranging from $-23.0\,\mu$g m$^{-3}$ to $-2.4\,\mu$g m$^{-3}$ with a mean of $-12.0\,\mu$g m$^{-3}$ (Figure 11b). The discrepancies between





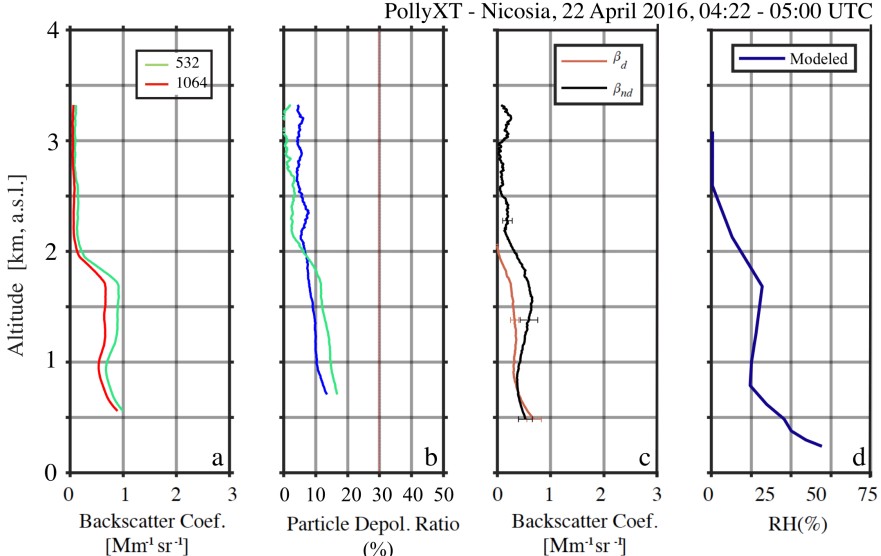

**Figure 10.** Daytime profiles of $\beta$ (532-, and 1064-nm wavelength) (a) $\delta^p$ (355- and 532-nm wavelength) (b) $\beta_d$ and $\beta_{nd}$ (c) determined by POLIPHON, as well as RH profiles from WRF−ARW model simulations over Nicosia at 04:00 UTC (d). The 355 nm channel of the lidar was discarded due to misalignment.

the two methods can be partly attributed to the assumptions used in POLIPHON: 1) constant S throughout the atmospheric column, 2) contribution in the coarse-mode only from depolarizing particles, and 3) the assumption of an externally-mixed aerosol. Assumptions used for the manipulation of the OPC measurements that can explain differences between the two methods are mainly related to the refractive index and the shape of the particles. The refractive index can notably influence the size distribution measured by OPC, inducing sizing uncertainties of up to 30% (Rosenberg et al., 2012; Granados-Muñoz et al., 2016). The refractive index used for calibrating $OPC_a$ , however, has a value of for n = 1.59, very close to literature values for Saharan dust (n = 1.56; Petzold et al., 2009). The difference between the refractive index values used for the calibration of OPCa and that used to for the retrieval of the LIDAR measurements is estimated to introduce a bias of 2% to the calculated mass concentration values.

Another source of the discrepancy between the mass concentrations determined by $OPC_a$ and the lidar is the limitation of the former to measure particles larger than a few tens of microns due to aerodynamic inlet loses (sedimentation and inertial deposition), resulting in an underestimation of 20% of the coarse-mode volume concentration (cf. yellow-green shaded area in Figure 3). To account for that, we corrected the $OPC_a$ measurements using the formula: $m_{\mathrm{OPC}} = m_{\mathrm{POLIPHON}} \frac{\int_{\mathrm{OPC}_a} dV/d\ln r}{\int_{\mathrm{POLIPHON}} dV/d\ln r}$. This correction significantly improved the agreement between the $OPC_a$ (green curve in Figure 11a) and the lidar measurements, constraining the bias range to $-11.1$ $\mu$g m$^{-3}$ and $8.8$ $\mu$g m$^{-3}$ which results in a decreased mean bias of $-1.1$ $\mu$g m$^{-3}$.





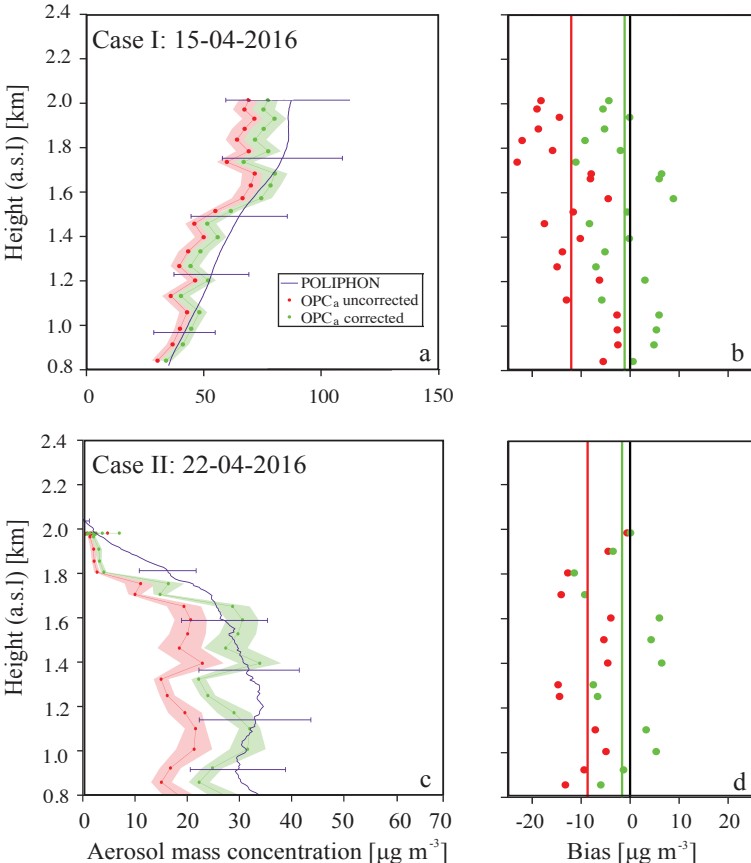

**Figure 11.** Aerosol mass concentration profiles for case study I and II (a, c) and the respected biases (b, d). In plots a and c, the blue solid lines show the mass concentration derived by the POLIPHON. The mass concentration measured by the $OPC_a$ is plotted in red with the red shaded area representing the uncertainties of the in-situ measurement. The green lines show the mass concentration from the $OPC_a$ corrected for the particles losses due to cut-offs. In plots b and d, the red dots show the biases between the values measured by the $OPC_a$ and the lidar ($OPC_a$ − lidar) before applying the correction, while the green dots are the biases after corrections. The red and green solid lines show the mean biases before and after correction, respectively.





The mass concentration profiles determined by the lidar and the $OPC_a$ measurements on 22 April 2016 (Figure 11c) also show a good correlation, with $R^2 = 0.72$. In terms of absolute values, the mass concentrations determined by the $OPC_a$ measurements (green line) are lower compared to those determined by the lidar observations for the entire range of altitudes, exhibiting biases in the range from $-14.7$ $\mu g$ $m^{-3}$ to $0.6$ $\mu g$ $m^{-3}$ with a mean value of $-8.7$ $\mu g$ $m^{-3}$ (Figure 11d). The

integrated volume size distribution measured by the sunphotometer in Nicosia (cf. Figure S3) showed that in this case the $OPC_a$ underestimates the coarse volume fraction by 48%. Upon correction, the mean bias decreases to $-1.6$ $\mu g$ $m^{-3}$ and, with the exception of one point at 1.8 km altitude, the mass concentration values from the $OPC_a$ lie within the calculated uncertainty resulting from the POLIPHON algorithm used to invert the lidar data (32%). At higher altitudes the mass concentration decreases drastically and $OPC_a$ measurements drop below the POLIPHON uncertainty limits.

Overall, the airborne in-situ and lidar observations are in good agreement both during the observation of a dense as well as of a weak dust event, after the necessary corrections for the $OPC_a$ measurements. In the case of the moderate dust event the volume concentration fraction that is not captured by the OPC range is small and so the corresponding correction is small. In contrast, during the weak dust event, the OPC misses almost 50% of the volume size distribution which introduces large

measurement ambiguities.

## 4   Summary and Conclusions

In this study we compare, for the first time to our knowledge, vertical profiles of the aerosol mass concentrations determined independently by an OPC on board a UAV and by remote sensing observations using data from a LIDAR and a sunpohotometer. The measurements were performed during two cases of dust events that occurred in the region of the Eastern Mediterranean

on 15 and 22 April 2016. During those days, the UAV flew up to ca. 2 km altitude with the OPC measuring the size distributions of sampled aerosol particles having radii in the range 0.15-5 $\mu m$, from which the aerosol mass concentrations were calculated. The same information was retrieved by concurrent lidar and sunphotometer measurements that were inverted using the POLIPHON method.

During the measurements on 15 April 2016 the dense dust layer extended from 2 to 4 km, while a mixture of dust and almost spherical particles was observed below 2 km. The mass concentration of the coarse-mode particles increased from 30 $\mu g$ $m^{-3}$, at ca. 0.8 km, to ca. 70 $\mu g$ $m^{-3}$, at ca. 1.8 km. Agreement between the in-situ measurements and the lidar observations retrieved with the POLIPHON method was very good ($R^2 = 0.8$), with the in-situ measurements lying within the POLIPHON uncertainty limits (38%), exhibiting a mean bias of $-12.0$ $\mu g$ $m^{-3}$ that can be mainly attributed to the difference in the cut-off

diameters measured by the two techniques. Corrections applied to account for this difference in the cut-off diameters further enhanced the agreement, decreasing the mean bias to $-1.1$ $\mu g$ $m^{-3}$.





In the measurements carried out on 22 April 2016, a sparse dust layer was observed between 0.8-2 km altitude during the morning hours. Information from the lidar measurements suggests that this layer was a mixture of desert dust particles with particles from another source. Despite that, however, agreement between the airborne in-situ and remote sensing measurements in this case was also very good ($R^2$ = 0.72). In terms of absolute values, the corrected mass concentrations measured by the OPC$_a$ were equal or lower than those derived from the lidar measurements for the entire range of altitudes and exhibited a mean bias of $-1.6$ $\mu$g m$^{-3}$. The concentrations measured by the OPC were within the calculated uncertainty of POLIPHON.

The measurements reported here indicate that unmanned airborne OPC measurements and lidar observations can provide reliable ways to determine coarse-mode aerosol mass concentration profiles in the atmospheric column, thereby bridging the gap between in-situ and remote sensing observations. Considering that both methods can provide dense datasets in a cost-effective manner and on a regular basis, this finding paves the way towards their systematic exploitation in climate models.

**Author contributions statement**

J.S., A.A. conceived the experiment, E.M., M.P., M.K., C.K., R.E. conducted the experiment(s), D.M. analysed the data, prepared the figures and wrote the manuscript. H.B. applied corrections to lidar profiles. G.B. supervised the work and contributed to the writing of the manuscript. P.K., I.B., A.T., E.M. and V.A. supported the data analysis procedure. All authors reviewed the manuscript.

*Data availability.* The sun photometer datasets used in this study are available at https://aeronet.gsfc.nasa.gov
The OPC datasets analysed during the current study are available from the co-author M.Pikridas (m.pikridas@cyi.ac.cy) on reasonable request.
The Polly$^{XT}$ LIDAR (http://polly.tropos.de) datasets analysed during the current study are available from the co-authors A. Ansmann (albert@tropos.de) and E.Marinou (elmarinou@noa.gr).

*Competing interests.* The authors declare no competing interests.

*Acknowledgements.* This project received funding from the European Union's Seventh Framework Program (FP7) project BACCHUS (Impact of Biogenic versus Anthropogenic emissions on Clouds and Climate: towards a Holistic UnderStanding) under grant agreement no. 603445, and the European Union's Horizon 2020 research and innovation program ACTRIS-2 (Aerosols, Clouds and Trace gases Research InfraStructure Network) under grant agreement No 654109.
The authors extend special thanks to Dr. Robin Lewis Modini from the Paul Scherrer Institute for assisting in the Mie Calculations.
PK acknowledges the funding of the Greek State Scholarship Foundation: IKY. Part of this project is implemented within the framework





of the Action "Reinforcement of Postdoctoral Researchers" of the Operational Program "Human Resource Development, Education and Lifelong Learning", and is co-financed by the European Social Fund (ESF) and the Greek State (NSRF, 2014-2020).



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
