# Peer review of "Vertical profiles of aerosol mass concentration derived by unmanned airborne in-situ and remote sensing instruments during dust events."

_Atmospheric Measurement Techniques, 2017_

## Referee Comment (RC1) · Anonymous Referee #2 · 1 Feb 2018

General comments

In general the presented material is very interesting and should be published, however I have some comments. Authors stated that mass concentration was derived from in-situ and remote measurements. I would like to see word "derived" in paper title. Article structure seems to be unsuitable for this journal, in my opinion the part related to description of individual cases is a little too long. The paper should rather be about technique then case studies. Authors show AOT and Angstrom Exponent time-series for two locations. Is it crucial for the presented methods or is it placed just to prove that

they compare the same air masses?

More specific comments regarding OPC measurements.

The used OPC has a measurement range much smaller than spectra obtained from CIMEL retrievals. It may be crucial in case of coarse mode. Authors show that 20% of coarse mode and most of fine mode is not covered by OPC spectrum. I can understand that applied correction (page 14, line 14) should mitigate this problem. I suggest fiting bimodal distribution to OPC data with boundary condition applied to CIMEL size distribution with zeros at the ends. Maybe it could help. Authors discussed uncertainties caused by refraction index of material used for calibration of OPC and index of real aerosol. Could they discuss uncertainties caused by aerosol shape? I can suppose that non-spherical aerosol may give different signals than spherical one used for calibration. Another question, in the manuscript is stated that authors used constant density (page 4, line 7) whilst in Appendix, section S1 they mentioned dust particle density and non-dust particle density. How was it really calculated, one density for the whole spectrum or different densities for different modes?

More specific comments regarding POLIPHON.

I understand that retrieval of mass concentration from POLIPHON method bases on assumption that coarse mode is dominated by large depolarizing particles. What happens when coarse mode is a mixture of polarizing and non-polarizing particles. In example mixture of dust and maritime aerosols? This is the case in second episode (page 13, line 4). Maybe, in case of second episode, authors should not apply correction to OPC measurements but to POLIPHON retrievals because assumptions of POLIPHON methods seems to be not fulfilled.

More specific comments regarding case studies.

In my opinion this section is a little too long compared to discussion and Mass Concentration Profiles section. In general Figures containing LIDAR signals, LIDAR quicklooks

and figures containing mass concentrations have different vertical scales. Unification of scales will help in quick comparison of results. I would also like to see time window of both LIDAR measurement, for LIDAR ratio and mass concentration estimation, as well as for UAV flight. What are colors in Figure 6. I can suppose that are altitudes of trajectory endpoints, it is not defined. Authors estimated LIDAR ratio (LR) around midnight. It is OK for Raman measurements. However during LR measurements and UAV flight different depolarization ratios are observed? Is it really the same aerosol? Increase of depolarization ratio (Figure 5) may suggest some changes. Could you please comment that. A few comments regarding trajectory analysis. I suppose that Figure 6 should prove that air mass originates from northern Africa. However, it passes over southern Italy, Greece and Turkey. Could you please comment possible influence of anthropogenic aerosol on your results. Height of trajectory may be large enough not to capture anthropogenic aerosols. However, it is not clear from the figure and text. The same for Figure 9, trajectories proving that it is dust start at the ground level. What about anthropogenic aerosol from Cairo or Alexandria?

More specific comments regarding mass concentration profiles

I would like to see discussion of uncertainties induced by shape of particles (OPC) and contribution of only polarizing particles to coarse mode (POLIPHON). It is mentioned but not discussed. Second thing. Corrections of OPC mass concentration by POLIPHON mass concentration and integrated volume size distributions makes sense when authors are sure that POLIPHON works well. In my opinion in second case study POLIPHON assumption is not fulfilled. That's why OPC correction is so large. I would rather extrapolate somehow (for example by fitting bimodal function) OPC size distribution and then compare OPC results with POLIPHON one. Regarding comparison of OPC and POLIPHON results. Statements that values are within error bars and provision of pretty large correlation coefficients looks great but could you give more sophisticated statistical analysis? In my opinion it is necessary especially in case of measurements taken in different locations. Could you provide tests for mean values

or for distribution of mass concentration. It may be done for whole population or for different altitude ranges.

---

## Referee Comment (RC2) · Anonymous Referee #1 · 8 Feb 2018

The work is of high interest, dealing with technological innovation and closure between different techniques. It is very well described and the use of English is of high level.

Just few minor comment:

Which is the time resolution and air flow used for the AE51?

The sun photometer inversion is valid for AOD equal to 0.1?

S1 Which is the original time resolution of the OPC data?

Technical details:

p 2 l 3: instrumental -> fundamental

p 8 l 7: define PBL here

Figure S1 why so different panels graphic?

―――――――――――――

---

## Author Comment (AC1) · 27 Mar 2018

We thank the reviewer for his time and effort. His/her comments have been very valuable for improving the quality of our manuscript. Below are our responses to all the points raised.

Point 1: Which is the time resolution and air flow used for the AE51?

The time resolution is 1 second and the air flow is 0.2 liters per minute. This information has now been added in the manuscript.

[Figure]

Point 2: The sun photometer inversion is valid for AOD equal to 0.1?

According to bibliography (Dubovik et al. 2000, Dubovik et al. 2001) retrieval of the particle volume size distribution was demonstrated to be adequate in practically all situations with AOT > 0.05. For low aerosol loading significant errors are induced in the retrieval of the single scattering albedo, the real and imaginary part of the refractive index. This information has been added in Section 2.6 of the manuscript.

Point 3: S1 Which is the original time resolution of the OPC data?

The time resolution of the OPC data is 1 second. This information has been added in Section 2.4 of the updated version of the manuscript.

Point 4: p 2 l 3: instrumental -> fundamental

The change has been applied.

Point 5: p 8 l 7: define PBL here

PBL is now defined in the manuscript.

Point 6: Figure S1 why so different panels graphic?

We understand the point of the reviewer here. As a response we have improved Figure S1 in the updated version.

References

1. Dubovik, O., et al. "Accuracy assessments of aerosol optical properties retrieved from Aerosol Robotic Network (AERONET) Sun and sky radiance measurements." Journal of Geophysical Research: Atmospheres 105.D8 (2000): 9791-9806.

2. Dubovik, O., B. Holben, T.F. Eck, A. Smirnov, Y.J. Kaufman, M.D. King, D. Tanré, and I. Slutsker, 2002: Variability of Absorption and Optical Properties of Key Aerosol Types Observed in Worldwide Locations. J. Atmos. Sci., 59, 590–608, https://doi.org/10.1175/1520-0469(2002)059<0590:VOAAOP>2.0.CO;2

[Figure]

[Figure]

**Fig. 1.**

---

## Author Response (AR1)

**RESPONSE TO REVIEWERS**

**RESPONSES TO POINTS RAISED BY REVIEWER #1**

We thank the reviewer for his time and effort. His/her comments have been very valuable for improving the quality of our manuscript. Below are our responses to all the points raised.

Point 1: Which is the time resolution and air flow used for the AE51?

The time resolution is 1 second and the air flow is 0.2 liters per minute. This information has now been added in the manuscript.

Point 2: The sun photometer inversion is valid for AOD equal to 0.1?

According to bibliography (Dubovik et al. 2000, Dubovik et al. 2001) retrieval of the particle volume size distribution was demonstrated to be adequate in practically all situations with AOT > 0.05. For low aerosol loading significant errors are induced in the retrieval of the single scattering albedo, the real and imaginary part of the refractive index. This information has been added in Section 2.6 of the manuscript.

**Point 3: S1 Which is the original time resolution of the OPC data?**

The time resolution of the OPC data is 1 second. This information has been added in Section 2.4 of the updated version of the manuscript.

Point 4: p 2 1 3: instrumental -> fundamental

The change has been applied.

Point 5: p 817: define PBL here

PBL is now defined in the manuscript.

Point 6: Figure S1 why so different panels graphic?

We understand the point of the reviewer here. As a response we have improved Figure S1 in the updated version.

**RESPONSES TO POINTS RAISED BY REVIEWER #2**

We thank the reviewer for his time and effort. His/her comments have been very valuable for improving the quality of our manuscript. Below are our responses to all the points raised.

Point 1: I would like to see word "derived" in paper title.

The title has been changed to:

*"Vertical profiles of aerosol mass concentrations derived by unmanned airborne in-situ and remote sensing instruments during dust events."*

Point 2: Article structure seems to be unsuitable for this journal, in my opinion the part related to description of individual cases is a little too long. The paper should rather be about technique then case studies.

The point raised by the reviewer is well taken. The methodology-technique part of the paper was extended by including parts that were previously included in the supplementary material, namely the sections:

- 1. Particle Mass Concentration Calculation from the OPCa Measurements
- 2. POLIPHON Method Error Estimation

Point 3: Authors show AOT and Angstrom Exponent time-series for two locations. Is it crucial for the presented methods or is it placed just to prove that they compare the same air masses?

We thank the reviewer for this point. Indeed we show this graph to prove that we compare the same air masses. To avoid confusion we added a clarification in an existing sentence in Section 3.1. The updated sentences reads as follows:

"This good correlation was further enhanced during the dust event cases (e.g. on 15 April 2016) when the relative contribution of the aerosol fine mode was minimized, which supports that a comparison of aerosol measurements at these locations is meaningful."

Point 4: I suggest fitting bimodal distribution to OPC data with boundary condition applied to CIMEL size distribution with zeros at the ends. Maybe it could help.

We understand the point of the reviewer here. We fitted lognormal distributions to the volume size distribution data of the  $OPC_a$  and, based on these curves, calculated the mass concentration. The results showed that in Case Study 1 the mass concentration calculated by the lognormal fits is in very good agreement ( $\pm$  5%) with the mass concentration calculated after correction based on the sun-photometer data (as described in the manuscript). However, in Case Study 2 the use of the lognormal fits did not improve the results and the differences between the two methods (lognormal fits and sun-photometer correction) reached up to 30 %.

Point 5: Authors discussed uncertainties caused by refraction index of material used for calibration of OPC and index of real aerosol. Could they discuss uncertainties caused by aerosol shape? I can suppose that non-spherical aerosol may give different signals than spherical one used for calibration.

We thank the reviewer for this comment. Yes, particle morphology can affect the sizing of the particles by OPCs as shown by Osborne et al. (2008). The following text was added in the updated version of the manuscript (see Section 3.2.3) in order to address this point:

"Regarding particle shape, the effect of non-sphericity on the particle sizing by light-scattering instruments having similar scattering angle range that of  $OPC_a$  (90°±60°) is within less than 20%, with a tendency towards undersizing (Osborne et al., 2008)."

Point 6: Another question, in the manuscript is stated that authors used constant density (page 4, line 7) whilst in Appendix, section S1 they mentioned dust particle density and non-dust particle density. How was it really calculated, one density for the whole spectrum or different densities for different modes?

We understand the confusion. We used different densities for the fine and coarse modes. However, since we do not show the comparison between the fine mode measurements, this information is redundant and confusing. Thus, we removed the sentence referring to the density of the fine mode in the Section 2.4 that was added in the main text as a response to point 1 of the reviewer (see above).

Point 7: I understand that retrieval of mass concentration from POLIPHON method bases on assumption that coarse mode is dominated by large depolarizing particles. What happens when coarse mode is a mixture of polarizing and non-polarizing particles. In example mixture of dust and maritime aerosols? This is the case in second episode (page 13, line 4). Maybe, in case of second episode, authors should not apply correction to OPC measurements but to POLIPHON retrievals because assumptions of POLIPHON methods seems to be not fulfilled. [... from another comment] The same for Figure 9, trajectories proving that it is dust start at the ground level. What about anthropogenic aerosol from Cairo or Alexandria?

We thank the reviewer for his very good points. His comments made us revisit Case Study 2 and modified the text in Section 3.2.2 accordingly:

"These relatively low  $\delta_{532}$  values indicate a mixture of Saharan dust with spherical continental/pollution particles. This is supported by the paths that the air mass follow between 1-2 km which originated from north-eastern Africa close to Cairo and Alexandria. The lidar ratio of  $40 \pm 7$  Sr, measured during the previous night (at a height where the signal is mostly free of noise; i.e. 1.2-1.4 km), agrees with the findings of Schuster et al. (2012) and Nisantzi et al. (2015) who reported respectively that  $S_{532} = 40 \pm$ 5 Sr and  $S_{532} = 47$  Sr for dust originating from eastern Sahara."

Point 8: In general Figures containing LIDAR signals, LIDAR quicklooks and figures containing mass concentrations have different vertical scales. Unification of scales will help in quick comparison of results. I would also like to see time window of both LIDAR measurement, for LIDAR ratio and mass concentration estimation, as well as for UAV flight.

Case Study I: The vertical scales of the quicklooks and the lidar retrievals are now the same (maximum altitude 8 km) and time windows were added to the lidar quicklook indicating the time spans of the Raman retrievals and the UAV flight (Figure 5 of the revised manuscript).

Case Study II: The plots were updated in a similar manner as in the previous case but with a maximum altitude of 3 km (Figure 8 of the revised manuscript).

We prefer to keep different vertical scales for the two case studies as we think that in that way it is more clear for the reader to grasp the atmospheric conditions during our measurements.

For the mass concentration profiles we used the maximum altitude that the UAVs flew (Figure 11).

Point 9: What are colors in Figure 6. I can suppose that are altitudes of trajectory endpoints, it is not defined.

The different colors are indicative of the altitudes of the different back-trajectories starting points. A legend was added to the plot for clarification.

Point 10: Authors estimated LIDAR ratio (LR) around midnight. It is OK for Raman measurements. However during LR measurements and UAV flight different depolarization ratios are observed? Is it really the same aerosol? Increase of depolarization ratio (Figure 5) may suggest some changes. Could you please comment that.

We thank the reviewer for this insightful comment. Indeed Fig. 5 shows that the depolarization ratio increases from 25% to 30% between the time of the Raman measurements and the time of the UAV flight (red rectangles). This implies that the dust layer initially was not pure but slightly mixed with another aerosol type, which explains why our Raman LR retrieval was 47 sr. The back-trajectory analysis showed that during both measurements the air masses arriving over our site, originated from the same region. Thus, we can safely assume that the dust particles originated from the same source.

Consequently, it is possible that the mean dust lidar ratio that was measured during night (Sd =  $47\pm10$  Sr) is valid and representative also for the day-time observations. However, former studies of Saharan dust in the region around Cyrpus (e.g. Nisantzi et al., 2015, Mamouri et al., 2016) report Saharan dust lidar ratios of Sd =  $53\pm6$  Sr. Taking into account the lidar ratio uncertainties ( $\pm10$  Sr), the Raman measured value of 47 Sr can therefore be safely used in the POLIPHON retrievals.

Point 11: A few comments regarding trajectory analysis. I suppose that Figure 6 should prove that air mass originates from northern Africa. However, it passes over southern Italy, Greece and Turkey. Could you please comment possible influence of anthropogenic aerosol on your results. Height of trajectory may be large enough not to capture anthropogenic aerosols. However, it is not clear from the figure and text.

The point of the reviewer is very well taken. Indeed, the altitude of the trajectory is too high (above the PBL) to have any direct influences from ground sources over Italy, Greece and Turkey. To make this clear we have clarified the discussion regarding the backtrajectories in Section 3.2.1:

"Backtrajectory analysis (cf. Figure 9) corroborated that this layer resulted from a Saharan dust event that originated in Algeria and traveled over Italy, Greece and Turkey before reaching Cyprus. Despite passing over polluted areas, the core of the dust layer remained pure (2.5-4 km, see analysis below) due to it high elevation (>2 km) throughout its path."

Point 12: I would like to see discussion of uncertainties induced by shape of particles (OPC) and contribution of only polarizing particles to coarse mode (POLIPHON). It is mentioned but not discussed.

These two issues have been addressed in our responses of points 5 and 7 of the reviewer (see above).

Point 13: Corrections of OPC mass concentration by POLIPHON mass concentration and integrated volume size distributions makes sense when authors are sure that POLIPHON works well. In my opinion in second case study POLIPHON assumption is not fulfilled. That's why OPC correction is so large. I would rather extrapolate somehow (for example by fitting bimodal function) OPC size distribution and then compare OPC results with POLIPHON one.

This point has already been addressed above (see responses to points 4 and 7).

Point 14: Statements that values are within error bars and provision of pretty large correlation coefficients looks great but could you give more sophisticated statistical analysis? In my opinion it is necessary especially in case of measurements taken in different locations. Could you provide tests for mean values or for distribution of mass concentration. It may be done for whole population or for different altitude ranges.

The reviewer has a good point here. We tested the differences between the means with a student t-test which in both cases showed that the difference between the in-situ and the remote sensing measurements is not statistically significant. The results of the tests are shown below. In order for our hypothesis (that the means of the two measurements are not statistically significant) to be valid the p values should be higher than the significance level 0.05. The results of the tests were added in the text in Section 3.2.3:

'Further statistical analysis between the lidar and the corrected  $OPC_a$  measurements showed that our hypothesis that the two observations refer to the same aerosol population is valid. To be more specific, the two-tailed T-test yielded a P-value of 0.70 (assuming equal variances), indicating that the differences between the mean values of the two types of observations are not statistically significant."

"Also in this case, the two-tailed T-test (assuming equal variances) yielded a P-value of 0.05 indicating marginal statistically insignificant differences between the means of the two types of measurements."

| 15/04/2016                           | in-situ           | lidar      |
|--------------------------------------|-------------------|------------|
| Mean                                 | 61.6794732        | 63.1364067 |
| Variance                             | 667.09617         | 347.776601 |
| Observations                         | 34                | 181        |
| Pooled Variance
Hypothesized Mean | 397.248647        |            |
| Difference                           | 0                 |            |
| df                                   | 213               |            |
| t Stat                               | -0.3910822        |            |
| P(T<=t) one-tail                     | 0.34806371        |            |
| t Critical one-tail                  | 1.65203888        |            |
| P(T<=t) two-tail                     | 0.69612742(>0.05) |            |
| t Critical two-tail                  | 1.97116389        |            |

**t-Test: Two-Sample Assuming Equal Variances**

The differences between the means are not statistically significant

**t-Test: Two-Sample Assuming Equal Variances**

| 22/04/2016                   | in-situ            | lidar       |
|------------------------------|--------------------|-------------|
| Mean                         | 20.1101707         | 24.95662895 |
| Variance                     | 135.155273         | 100.7737154 |
| Observations                 | 19                 | 167         |
| Pooled Variance              | 104.137129         |             |
| Hypothesized Mean Difference | 0                  |             |
| df                           | 184                |             |
| t Stat                       | -1.9615539         |             |
| P(T<=t) one-tail             | 0.02566152         |             |
| t Critical one-tail          | 1.65317709         |             |
| P(T<=t) two-tail             | 0.05132305 (>0.05) |             |
| t Critical two-tail          | 1.97294054         |             |

**The differences between the means are not statistically significant**

**Vertical profiles of aerosol mass <del>concentrations observed during</del> <del>dust events concentration derived</del> 
[revised manuscript text omitted]

|--------------------------------------------------------------------------------------------------------------------------------|-------------------------|-----------------------------|
| Dust depolarization ratio                                                                                                      | $\delta_d$              | $31\pm4\%$                  |
| Non-dust depolarization ratio                                                                                                  | $\delta_{nd}$           | $5\pm1\%$                   |
| Dust lidar LIDAR ratio                                                                                                         | $\mathbf{S}_d$          | $47\pm10~\mathrm{sr}$       |
| Non-dust <del>lidar_LIDAR</del> ratio                                                                                          | $\mathbf{S}_{nd}$       | $60\pm10~{ m sr}$           |
| Dust particle density                                                                                                          | $ ho_d$                 | $2.6 \pm 0.6 \ g \ cm^{-3}$ |
| Non-dust particle density $\rho_{nd}$ 1.5 $\pm$ 0.3 g cm -3 (Hess et al., 1998) Dust conversion factor (15.04.2016) | $\overline{v_c/\tau_c}$ | $0.67\pm0.05\times10^{-1}$  |
| Non-dust conversion factor (15.04.2016)                                                                                        | $\overline{v_f/	au_f}$  | $0.24\pm0.018\times10$      |
| Dust conversion factor (22.04.2016)                                                                                            | $\overline{v_c/\tau_c}$ | $0.81\pm0.04\times10^{-1}$  |
| Non-dust conversion factor (22.04.2016)                                                                                        | $\overline{v_f/	au_f}$  | $0.14\pm0.019	imes10$       |

**3 Results and Discussion**

15

5

10

**3.1 Homogeneity of Aerosol Measurements Properties over the Measurement Sites**

Given the proximity (6.5 km) of the ground (at CAO) and the airborne in-situ observations (at Orounda), as well as the absence of any strong pollution sources in the region, the measurements were considered to correspond to the same air parcel in terms of atmospheric composition. The third measurement location (Nicosia) was situated 35 km away from the airfield. As suggested

from by the comparison of sunphotometer measurements at Nicosia and CAO, however, all locations were influenced affected by the same air masses with minor local influence which was influence from local emissions that were mostly trapped within the PBLPlanetary Boundary Layer (PBL).

Figure 4 shows the AOT500 and the  $Å_{440-870}$  measured by the sunphotometers in Nicosia and at CAO from 13 to 24 April 2016 when concurrent measurements were performed at the two locations. Overall, the temporal variability of these two parameters observed at Nicosia was very similar with the respective measurements at CAO, exhibiting correlations coefficients of 0.89 and 0.87 for AOT500 and  $Å_{440-870}$ , respectively. This good correlation was further enhanced during the dust event cases (e.g., on 15 April 2016), when the relative contribution of the aerosol fine mode was minimized, supporting that a comparison of aerosol measurements at these locations is meaningful. In terms of absolute values, AOT500 was 15-50% higher at Nicosia compared to CAO, even during the cases with the dust events, when coarse particles dominated. These higher values at Nicosia are mainly due to the altitudinal difference between the sites (Nicosia is at an altitude of 190 m whereas CAO at 535 m above

sea level) and the contribution of the local aerosol sources to the total aerosol burden. This was further justified by the higher  $\mathring{A}_{440-870}$  measurements at Nicosia which signify the presence of small aerosol particles from anthropogenic sources.

**3.2 Comparison of the Mass Concentration Measurements**

A total of 6 UAV flights with  $OPC_a$  onboard were performed during the entire campaign. However, only 2 fulfilled all the necessary requirements for comparison with the lidar\_LIDAR observations. Those requirements are that 1) there are simultaneous measurements of lidar\_LIDAR and  $OPC_a$ , 2) there are cloud-free conditions or clouds are above 7-8 km altitude so that the lidar\_LIDAR retrievals can be made, 3) there is enough dust loading, 4) there is availability of AERONET data, and 5) the airborne in-situ measurements were performed before the full development of the PBL. All these requirements were fulfilled during the measurements on 15 April 2016 and 22 April 2016, which are analyzed below.

**3.2.1 Case Study I: 15 April 2016**

5 The atmospheric situation over South-Eastern Europe on 15 April 2016 was dominated by a high-pressure system resulting in mostly cloud-free conditions over Cyprus. A dust event of moderate intensity was observed, resulting in an average  $AOT_{500}$ value of 0.4 over Nisosia and CAO (cf. discussion in section Section 3.1 and Figure 4). Figure 5 shows the lidar-LIDAR